# Body Image and Physical Activity and the Diet of Polish Youth Aged 15–18

**DOI:** 10.3390/ijerph20043213

**Published:** 2023-02-12

**Authors:** Ewelina Łebek, Andrzej Knapik

**Affiliations:** 1Doctoral Studies, School of Health Sciences in Katowice, Medical University of Silesia in Katowice, Medyków 12, 40-752 Katowice, Poland; 2Department of Adapted Physical Activity and Sport, School of Health Sciences in Katowice, Medical University of Silesia in Katowice, ul. Medyków 12, 40-752 Katowice, Poland

**Keywords:** adolescent, body image, physical activity, eating behavior

## Abstract

Adolescence is a crucial stage in the biological, psychological, and social development of humans. During this period, perceptions of one’s body and behaviors are formed. The aim of the study was to investigate body image (BI) and its relationship with physical activity and selected eating habits in adolescents. The study included 312 people (102 girls (32.69%) and 210 boys (67.31%)) aged 15–18. As many as 40% of the girls and 27% of the boys reported feeling dissatisfied with their body mass. BI was negatively perceived by the adolescents, with girls being more critical than boys. Lack of acceptance of one’s body mass negatively affects overall BI in girls, and only the functional aspects in boys. Negative perception of body mass in girls does not motivate them to increase physical activity but rather causes them to adopt dietary restrictions.

## 1. Introduction

Adolescence is a period in which enormous changes in all spheres of human development occur. It is a time of shaping one’sidentity andis based primarily on biological maturation. It is associated with changes in social relations as well as in intellectual and mental functioning [1]. At this time, one’s senses of autonomy and coherence are shaped and one’s personal values and principles are defined [2]. An important part of this process is body image (BI) [3]. BI is a multidimensional construct encompassing individual perception and attitude towards one’s body, its physical features, and its functional aspects [4]. BI may change over time and is influenced by personality traits as well as cultural, social, and historical factors [5,6,7]. Due to the link between the perception of one’s body and attitudes towards self and others, BI is correlated with self-esteem [8].

Changes occurring during adolescence, the key differentiating factor of which is gender, may strongly affect the perception of one’s corporeality [9]. During this time, boys tend to gain a larger amount of lean body mass, whereas in girls, alean body mass and adipose tissue mass increase is observed [10]. The effects of these changes are confronted with promoted “ideals”. Nowadays, mainly through the media, women with a slim or very slim figure and men with large muscle mass and good physical fitness are promoted as the standards of beauty. Perceiving oneself as someone who does not meet these standards may be the source of many emotional problems and may also impinge on the behavior of adolescents [11,12,13]. These relations can be very important, in terms of both physical and mental health. There is no empirical research that provides a comprehensive analysis of these very important lifestyle characteristics of youth. The authors decided to investigate BI and its relationships with physical activity and selected dietary behaviors of adolescents.

## 2. Materials and Methods

The study included 312 individuals aged 15–18. The sampling was purposeful—the respondents were high school students (from Southern Poland). Consent to conduct the survey was obtained from the principals and teachers at the schools in which it was administered after they reviewed the questionnaire. Participation in the survey was voluntary. The questionnaire was administered using a paper–pencil method. The surveyed group included 102 girls (32.69%) and 210 boys (67.31%). The age distribution was as follows: 15 years: *n* = 105 (33.65%); 16 years: *n* = 88 (28.21%); 17 years: *n* = 76 (24.36%); 18 years: *n* = 43 (13.78%).

The original research tool, the questionnaire, consisted of several questions. These concerned information on height and body mass, based on which the BMI was calculated for each respondent. The respondents were also asked about acceptance of their body mass. The following response options were provided: no; no opinion; yes.

Body esteem was assessed using the Body Esteem Scale (BES) [14]. The questionnaire comprises 35 parts or functions of the respondents’ bodies rated on a Likert scale: 1—strong negative feelings; 2—moderate negative feelings; 3—no feeling one way or the other (neither negative, nor positive); 4—moderate positive feelings; 5—strong positive feelings. The scoring system allows one to calculate mean body self-esteem score in three domains. For women: Sexual Attractiveness (SA), Weight Concern (WC), and Physical Conditions (PC). For men: Physical Attractiveness (PA), Upper Body Strength (UBS), and Physical Conditions (PC).

The physical activity of the respondents was assessed using a modified Baecke physical activity questionnaire [15]. The full version of the questionnaire consists of 16 statements to which the respondents select a response. With a score assigned to each response item, it is possible to calculate, using an algorithm provided by the questionnaire’s authors, physical activity scores in three areas: work, sport, and leisure time (excluding sport). In addition, the summary activity score is obtained by adding up the scores in all three domains. Due to the relative homogeneity in terms of workload (the respondents were students), the authors decided to omit the work index. The question about sport asks if the respondent engages in a sport, the kind of sport played by the respondent, the number of hours per week, and the number of months per year that the respondent plays the sport. This section also contains a question on the respondent’s self-evaluation of physical activity in comparison to other people of the same age. As for the questions on leisure time, they concern leisure activities, including cycling and walking. Due to its clarity and simplicity, this questionnaire is used in studies. It was also validated using biochemical methods [16].

The respondents were also asked questions regarding physical education classes at school. The questions concerned:-Participation in classes: (1—I do not attend class; 2—I often skip class; 3—I sometimes skip class; 4—I rarely skip class; 5—I always participate in class)-Attitude towards class: (1—strongly negative; 2—rather negative; 3—neutral; 4—rather positive; 5—strongly positive).

Eating behaviors in adolescents were analyzed with the use of original statements, each of which had a set of response items with an assigned score. The respondents were asked about:(1)Diet quality control—avoiding sweets, crisps, fastfood, sugar-sweetened fizzy drinks:(1) no—I eat what I like; (2) rarely; (3) sometimes; (4) often; (5) yes(2)Number of meals:(1) one or two (without “grazing”); (2) one or two (with “grazing”); (3) three or more (without “grazing”); (4) three or more (with “grazing”)(3)Have they ever followed a dietto lose weight:(1) no; (2) yes, but for less than a week; (3) yes, for a long period of time.

### Statistical Analysis

Internal consistency was calculated using Cronbach’s alpha coefficient (CA). Descriptive statistics of the participants were performed. Means, SDs, and medians were calculated. To calculate differences, the following nonparametric statistics were used: Mann–Whitney U Test and Kruskal–Wallis one-way ANOVA. The correlations between the BES domains and the analyzed variables were calculated using Pearson correlation. The level of statistical significance was set at less than 0.05. All statistical analyses were performed with STATISTICA ver.13.3.

## 3. Results

A preliminary internal consistency analysis of the BES showed optimal Cronbach’s alpha values. For the specific domains, they were as follows: girls: SA: CA = 0.87; WC: CA = 0.91; PC: CA = 0.90. Boys: PA: CA = 0.89; UBS: CA = 0.88; PC: CA = 0.91.

Of the surveyed respondents, as many as 40% of girls and 27% of boys reported feeling dissatisfied with their body mass. As for boys, 27% of them reported lack of acceptance, 10% had no opinion, and 63% accepted their body mass. The results of gender comparison indicated a difference regarding the lack of acceptance between genders (U = 7564, *p* < 0.05).

Age was not a differentiating factor in terms of the scores obtained in the BES domains by boys or girls (*p* > 0.05). It was found that age did not differentiate between the scores obtained by girls or boys in the BES domains.

Gender comparison showed differences in terms of the analyzed behaviors (physical activity and dieting). The exceptions were physical activity during leisure time and diet quality control (Table 1). The differences between boys and girls in terms of summary physical activity resulted from increased physical activity levels in boys.

The relations between the analyzed variables (age, BMI, activity, body mass acceptance, and eating behaviors) and self-esteem were also investigated.

There were differences found between the girls who accepted their body mass and those who did not. They were found in all three BES domains: SA-U = 584.5, *p* < 0.001; WC-U = 226.5, *p* < 0.0001; PC-U = 533.5, *p* < 0.0001. The girls who accepted their body mass obtained higher scores in all domains. No difference was observed in boys in the PA domain. There were, however, differences in the UBS (U = 2793.5, *p* < 0.01) and PC (U = 2791.5, *p* < 0.01) domains (see Figure 1 below). The boys who accepted their body mass obtained higher scores in both domains.

In terms of behaviors, the authors found a positive correlation between the girls’ attitudes towards P.E. class and the PC domain and a negative correlation between dieting and the WC domain. No correlation was found between the PA domain and the analyzed variables in boys. The domains UBS and PC correlated with sport and summary activity (Table 2).

The analyzed elements of behaviors—physical activity and behaviors related to dieting—were also compared and the grouping variable was body mass acceptance (no/yes). In girls, the variable was a differentiating factor for leisure time activity (U = 762.5, *p* < 0.05), summary activity (U = 785, *p* < 0.05), and dieting (U = 514.5, *p* < 0.0001) (see Figure 2 below). No difference was observed for boys.

## 4. Discussion

The descriptive statistics analysis indicated a rather negative body perception among the respondents. Both means and medians in the BES domains ranged from indeterminate tomoderately positive feelings. Slightly higher self-esteem was observed in boys. This seems typical—it has already been noted in earlier studies that women more often display negative perceptions of their bodies [17]. A fairly critical assessment of BI indicates high expectations of youthin relation to promoted standards. This is a very important observation, as “body image is a crucial part of adolescent development for teens all over the world” [18]. Due to the lack of age-related differences, the study group can be treated as relatively homogeneousin terms of BI.

The presented results are lower than those obtained in the Polish study conducted among the residents of Szczecin in 2015 [19]. Perhaps the differences are due to the varying sample sizes—in the presented study, the group was three times larger. They may also result from the selection of respondents—first-year students also participated in the study. Self-perception is significantly influenced by external information. Thus, it can be assumed that an earlier change in environment in people who enrolled at a university may have had a positive effect on the perception of their body image.

Boys obtained higher BMI values than girls. However, there were statistically significant differences regarding the lack of acceptance of one’s body mass—a larger percentage of girls did not accept it. This confirms the influence of the prevailing “cult of thinness” as a standard for girls and women [20]. Body image dissatisfaction (BID),understood as a perceived discrepancy between one’s actual body and the desired ideal body, is a common phenomenon worldwide. BID affects a more significant percentage of overweight or obese people than underweight people [21]. Compared to girls, higher BMI in boys is not as often associated with adiposity but rather with muscle mass, which is widely considered one of the attributes of male corporeality [22]. The shaping of this attributethrough physical activity explains, to some extent, the differences between girls and boys in terms of sports activity and, consequently, summary activity (Table 1). These differences should be considered typical, as confirmed by the results of studies from many other countries [23,24,25]. The explanation for these differences is multidimensional. In addition to biological determinants, one should also take into account various psychosocial, cultural, and economic factors—mainly concerning preferences—that also play a role. Differences in the presented activity are, to a certain degree, related to participation in physical education lessons and attitudes towards the subject, in which case, more favorable results were also noted for boys. The observations presented here are in line with previous studies conducted by the authors [26]. One explanation for these differences may be higher expectations among girls regarding the organization, quality, and attractiveness of the lessons [27]. It should be noted, however, that according to the applied scoring system, the youth generally reported a positive attitude and participation in the classes. These findings oppose some previous reports [28]. It seems as though explaining these discrepancies requires the widest possible multicenter research, which would also provide an empirical explanation of the determinants of attitudes toward these classes.

Gender was found to be a significant factor in differentiating eating behaviors. While there were no differences in terms of quality of diet, boys reported eating more meals and were significantly less likely to follow a special diet to lose weight. One plausible explanation for these differences is the higher energy requirements of boys, due to both body structure and statistically increased levels of activity. In turn, the lack of differences regarding the PA domain, activity, and eating behaviors between the boys who accept their body weight and those who do not accept it indicates that this factor does not play a significant role when it comes to aspects of their appearance as well as behaviors. In contrast, lack of acceptance of body mass has an impact on boys’ self-assessment of functional aspects of BI (UBS and PC). In girls, acceptance of one’s body mass plays a very important role. Lack of acceptance results in lower BI in all the analyzed domains and more frequent adoption of dietary restrictions.

Many factors influence the eating behaviors of young people. These include food availability, individual preferences, family patterns, peer influence, personal and cultural beliefs, and the influence of mass media and BI [29]. The presented results indicate that BI has a greater influence on girls. The coincident lower level of physical activity in these girls may be a cause for concern. Improper use of dietary restriction strategies may lead to the development of eating disorders, which are a major health problem [30,31]. Negative perception of one’s own body mass may also be a predictor of activity avoidance that leads to kinesiophobia [32].

The importance of BI in the development of young people does not pertain only to health-related behaviors—physical activity or dieting—but also reflects attitudes and interactions with others. The stereotype of linking one’s physical attractiveness with positive traits is not only characteristic of Western culture but is also common throughout the world [33]. According to the sociocultural approach, cultural values shape one’s perceptions of others within the framework of prevailing beauty standards and expectations associated with them. One’s self-perception is influenced by how one is perceived by other people [34]. In terms of interactions with others, an empirical study showed that people with positive body image tend to surround themselves with people who feel positive and accepting of their own bodies [35]. Therefore, it is desirable to create targeted educational programs aimed not only at young people but also at their parents. These programs should take into account the role and importance of BI in young people for their physical and mental well-being, and should promote optimal health behaviors regarding dieting and physical activity.

## 5. Study Limitations

The cross-sectional nature of the study, its limited scope, and the number of participants pose some limitations to this study. Further, the test method used excludes the direct measurement of height and weight, which results in a certain error in the BMI. The methods used for the anthropometric measurement were not objective. The students self-reported their weight and height in the questionnaire.However, the importance of BI in adolescents suggests the importance of conducting multicenter studies to confirm the observations and consequent conclusions presented in them. One limitation of this study was the method of measuring physical activity using the Baecke questionnaire. It would have been more accurate to measure it with an accelerometric device; however, due to the choice to conduct a survey-based study, the authors opted to administer standardized questionnaires that are widely used and accepted in survey studies.

## 6. Conclusions

BI is perceived rather negatively by adolescents, with girls being particularly critical of WC. Lack of acceptance of one’s body mass negatively affects all domains of the BES in girls, whereas in boys, it affects functional aspects (UBS and PC). Negative perception of body mass in girls does not motivate them to increase physical activity but rather causes them to adopt dietary restrictions. Dietary restrictions in girls are negatively correlated with WC. Positive correlations of physical activity in boys relate to PA and UBS.

## Figures and Tables

**Figure 1 ijerph-20-03213-f001:**
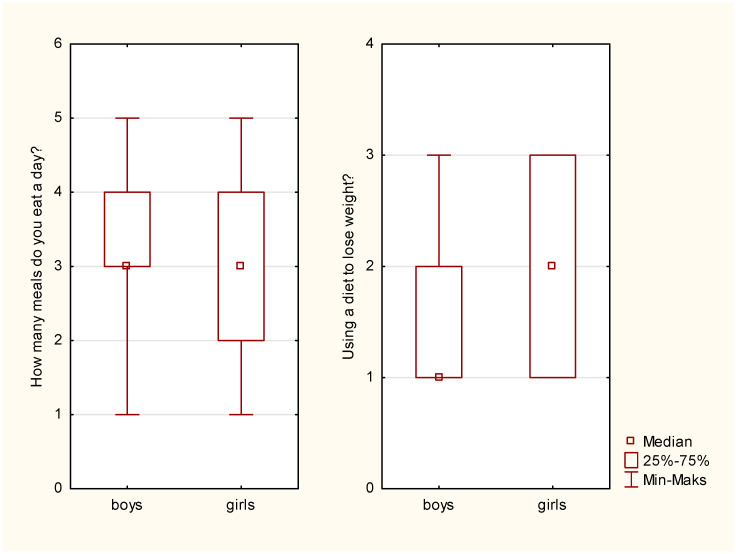
Number of meals and dieting—comparison by gender.

**Figure 2 ijerph-20-03213-f002:**
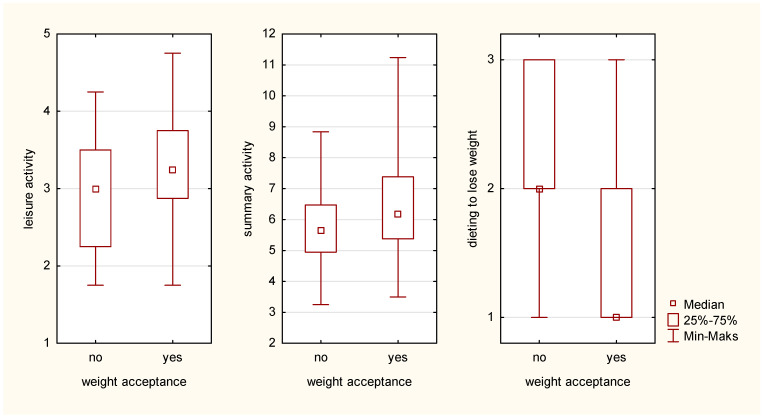
Comparison of girls not accepting their body mass and girls accepting their body mass.

**Table 1 ijerph-20-03213-t001:** Descriptive statistics of the analyzed variables and comparison by gender.

Variable		Girls		Boys		Girls–Boys: *p*
		Mean (SD)	Median	Mean (SD)	Median	
age		16.01 (1.08)	16	16.27 (1.03)	16	*
BMI		20.44 (3.17)	20.22	22.15 (3.81)	21.38	**
BES	SA	3.74 (0.78)	3.73			
	WC	3.35 (1.02)	3.40			
	PC	3.72 (0.82)	3.78	3.98 (0.72)	4.80	
	PA			3.96 (0.69)	4.09	
	UBS			3.96 (0.75)	4.00	
physical activity	sport	3.01 (1.18)	2.71	3.57 (1.41)	3.49	**
	leisure time	3.14 (0.65)	3.00	3.06 (0.73)	3.00	
	summary	6.15 (1.47)	5.95	6.63 (1.77)	6.49	*
physical education class	participation	3.98 (1.11)	4.00	4.19 (1.17)	5.00	*
	attitude	3.81 (1.17)	4.00	4.09 (1.10)	4.00	*
diet	quality control	2.78 (1.37)	3.00	2.73 (1.40)	3.00	
	number of meals	2.87 (1.03)	3.00	3.20 (0.93)	3.00	**
	dieting	1.86 (0.80)	2.00	1.53 (0.82)	1.00	***

* *p* < 0.05; ** *p* < 0.001; *** *p* < 0.0001.

**Table 2 ijerph-20-03213-t002:** Correlation between the analyzed variables and the BES scores.

Variable		Girls			Boys		
		SA	WC	PC	PA	UBS	PC
BMI			−0.297	−0.198	−0.190	−0.197	−0.409
physical activity	sport			0.138	0.326	0.330	
	leisure time		0.248	0.147	0.172	0.164	
	summary		0.199	0.171	0.332	0.333	
physical education class	participation					0.125	
	attitude		0.207	0.185			
diet	quality control	−0.203	−0.208		0.144		
	number of meals						
	dieting		−0.414			−0.170	

Statistically significant coefficients were included: *p* < 0.05.

## Data Availability

The data presented in this study are available on request from the corresponding author. The data are not publicly available due to the risk of plagiarism.

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
