# Peer review of "Body Image and Physical Activity and the Diet of Polish Youth Aged 15–18"

_ijerph, 2023, doi:10.3390/ijerph20043213_

Round 1

Reviewer 1 Report

The study research elaborates an original but not novel hypothesis. The work is clear and easy to read, although the authors could have developed more their discussions based on the data obtained.
Certain mistakes and information omissions are observed, however, once solved, it will improve the quality of the study.

Please, read carefully the comments explained below:

1-Line 35: tend to gain a larger amount of lean body mass...

2-Line 42: you state that: There is little empirical research, but I suppose that there will be anything. Try to reference some examples. If you cannot find it, you should change the sentence adding: There is no empirical research…

3-Line 47: Try better: 312 individuals instead of people.

4-Line 56-57: This is not an objective method to assess anthropometry. You can highlight it in the limitations.

5-Line 67: This can be useful, but it is not an objective method of measuring physical activity, such as accelerometry devices. You can highlight it in the limitations.

6-Line 75: Please, remove one "the".

7-Line 80: “often used in studies”. Please, put more examples of studies that use this method.

8-Line 105: “a p-value<0.05 was adopted”. Suggestion. maybe is better to write: setting the overall significance level as α = 0.05.

9-Line 115: You have explained in the methods all the statistical methods you will apply. For a better understanding, it is better to specify the result obtained by applying the used method. Or, if you are not going to say the used method, you can put the U or H value with the p-value. For example: (H statistic = xx.x; p-value < x.xxx) or (U statistic = xx.x; p-value < x.xxx). Please, apply it in all the cases that appear in the text.

10-Table 1. “(1)-(2): p". Tables should be self-explanatory. In that case, it could be better if you write: sex comparison or: p-value. On the other hand, if you decide to leave “(1)-(2): p" you have to explain it.

11-Figure 1: Please, replace the commas with points in the ordinate axis values. Do it in the second Figure. Besides, you only have 2 figures but you appoint your second figure as "Figure 3", change this error to all your text.

12-Line 128-129: This has already been said in the methods. Go straight to reveal your results.

13-Table 2: Perhaps, it could be also interesting if you describe that Table 2 values are the r2.

14-Line 126: the authors found positive correlations... or a positive correlation..., but not positive correlation. The same in Line 137: a negative correlation...

15-Line 142: Does it mean that you have included only statistically significant correlations? or maybe all the correlations are statistically significant?

16-Line 145-146: Please, follow the same advice as displayed in review number 9.

17-Line 166: “Naturally”. This expression is a preconceived opinion and it is a statement without any support. If it is something that is highly supported by other studies, please support it with bibliography.

18-Line 178,179,180: Please, support this sentence with bibliography.

19-Line 190: which would also provide an empirical explanation...

20-Line 192: Gender was found to be a significant factor in differentiating eating behaviors.

21-Line 193: While there were no differences in terms of the quality of the diet...

22-Line 194,195,196: I agree, but in that case, you can state it because at the ages here studied (15-18) (after the growth spurt in boys), boys tend to be larger in height and weight, and they have more fat-free mass. Therefore, due to the relationship between body weight and energy expenditure, and due to at this age males are bigger than girls, their energetic requirements are higher. However, with the same body size and physical activity, males and females tend to expend the same energy.

Alike, they tend to perform more physical acivity. However you should improve this sentence following that information and supporting all with references.

23-Line 198: those who do not indicate that this factor...

24-Line 219: In this line, you talk about "studies" but you only reference one. Search for other examples, or else, write: “an empirical study (35)”…

Another option could be: In terms of interaction with others, Alleva and colleagues (2021) reveal in an empirical study that people with positive…

25-Line 226: Please, here you should include some of the limitations displayed in these reviews (Review 4 and 5).

26-Line 228-229: The importance of BI in adolescents suggests conducting a multicenter study to confirm... or ... conducting multicenter studies to confirm...

27-Line 231: The BI is perceived rather negatively by adolescents...

Author Response

Please write see the attachment

Reviewer 2 Report

The passive voice is used quite frequently throughout; consider editing to change to active voice (instead of "this was done," "we did this").

May want to make sure you use the language cisgender male and cisgender female, as it doesn't appear that any gender diverse individuals were included in this study?

I don't see any demographic data for this study.  Was that collected?  That can be important in controlling for confounders or interpreting the results through a certain lens.

It would be helpful for the authors to indicate more clearly in the introduction about how this is a novel study, looking at a different question from prior research studies.

I would warn about making conclusions that are too sweeping and that imply causation and instead make sure the language indicates that the findings mean certain causes are possible.  For example, the paragraph starting with line 192 uses strong language that makes assumptions about reasons for the results, however the data don't support a causative relationship.

Similarly, the conclusion makes it seem as though the study has proven the connections between things like weight concerns and body image--but the methods only allow for associations.  Would be careful with the language used there.

Round 2

Reviewer 1 Report

I have observed that improvements have been introduced.
Considering these improvements, I have no more revision to contribute.
For my part, the paper can be published.